# The Microscopic Morphology of Mouthparts and Their Sensilla in the Mycophagous Ladybeetle *Illeis chinensis* (Coleoptera: Coccinellidae)

**DOI:** 10.3390/insects15010046

**Published:** 2024-01-09

**Authors:** Ke Wang, Yuanyuan Lu, Ming Bai, Yuanxing Sun, Yanan Hao

**Affiliations:** 1Biocontrol Engineering Laboratory of Crop Diseases and Pests of Gansu Province, College of Plant Protection, Gansu Agricultural University, Lanzhou 730070, China; wang_ke00@126.com (K.W.); sunyx1988@126.com (Y.S.); 2Key Laboratory of Zoological Systematics and Evolution, Institute of Zoology, Chinese Academy of Sciences, Beijing 100000, China; luyuanyuan@ioz.ac.cn (Y.L.); baim@ioz.ac.cn (M.B.); 3Northeast Asia Biodiversity Research Center, Northeast Forestry University, Harbin 150040, China

**Keywords:** fine morphology, mouthparts, sensilla, *Illeis chinensis*, scanning electron microscopy

## Abstract

**Simple Summary:**

*Illeis chinensis* feeds on various powdery mildews. It plays a vital role in spreading spores. This paper provides a detailed description of the fine morphology of each of the mouthparts of *I. chinensis* along with a careful identification and classification of its diverse sensilla types. Specifically, the differences between *I. chinensis* and predatory ladybeetles are compared, and the functions of each of the mouthparts, as well as their different kinds of sensilla, are discussed.

**Abstract:**

The morphological diversity of insect mouthparts is closely related to changes in food sources and diets. Research into the structures of insect mouthparts may help to establish a fundamental basis for a better understanding of insect feeding mechanisms. In this study, we examined the fine morphology of the mouthparts of *Illeis chinensis* using scanning electron microscopy. We paid particular attention to the types, quantities, and distribution of sensilla on the mouthparts. Our results showed that the basic components of the mouthparts of *I. chinensis* are the same as those in other lady beetles, i.e., the labrum, mandible, maxillae, labium, and hypopharynx. We also found structural specialization indicating adaptation to fungal feeding. On the mouthparts, there are eight kinds of sensilla and two kinds of glandular structures, including sensilla chaetica, sensilla basiconica, sensilla styloconica, sensilla coeloconica, sensilla campaniformia, sensilla placodea, sensilla digitiformia, Böhm bristles, perforated plates, and cuticular pores. This is the first time that sensilla digitiformia has been reported in ladybirds. Finally, variations in mouthparts among ladybirds with differing diets, as well as the putative functions of each of the mouthparts and sensilla, were discussed. This research can provide a reference for understanding the functions of the mouthparts in ladybird feeding behavior and thereby contribute to the development of precise insect behavior regulation and management strategies.

## 1. Introduction

Insect mouthparts, as highly integrated structural units [1], exhibit significant modifications in type and morphology due to variations in feeding objects and behaviors [2,3]. The morphological diversity of mouthparts is closely related to changes in food sources and feeding behaviors [4,5]. For instance, specialized mouthparts adapted for pollen or nectar feeding have been observed across various beetle taxa, as well as contrasting mating behaviors, often coupled with the different feeding guilds [6]. In the case of the hematophagous horn fly, details of the lacerating and penetrating structures on mouthparts employed during hematophagy have been observed, and the different sequential feeding positions assumed by mouthparts have been recorded using a specific methodology which allows researchers to highlight the functioning of microstructures associated with hematophagous feeding [7]. The study of mouthpart structures therefore provides a foundation for further understanding of feeding mechanisms [8,9,10]. In addition, adaptations in mouthpart morphology can reveal the interplay between feeding adaptations and mating behavior [6].

The sensillum is the main allelopathic organ of the mouthparts, and sensilla are widely distributed on the mouthparts as well as antennae. They exhibit diverse functions and play crucial roles in foraging and the mate-searching process [11,12,13]. Aggregations and types of sensilla on the mouthparts are directly related to feeding habits. Investigation of their distribution on different mouthparts may provide insight into their function and utility in the location of prey. Research on the functional morphology of sensilla may also advance our knowledge of insect feeding behavior and thereby contribute to their application in biological control programs [13].

Numerous studies have been conducted to examine the morphology of mouthparts in Coleoptera insects such as Coccinellidae [3,13], Nitidulidae [14], Cerambycidae [15,16,17], Carabidae [18], and Chrysomelidae [19]. However, almost all previous studies have focused primarily on predatory and phytophagous species; there have been few studies on mycophagous insects [1]. The family Coccinellidae encompasses phytophagous, mycophagous, and predatory species [20]; to date, mouthpart morphology has been extensively studied in predatory species [2,3,13,21] but not in mycophagous species.

*Illeis chinensis* is a common species in Psylloborini of Coccinellinae; it is distributed across China, the United States, Vietnam, and other regions [22]. This species exhibits high fecundity and a prolonged life cycle and feeds on various powdery mildews. Furthermore, it plays a vital role in spreading spores [23,24]. As the main feeding organs, mouthparts play crucial roles in feeding and spore spreading. In this study, the fine morphology of the mouthparts in mycophagous ladybirds was examined by scanning electron microscopy for the first time, and the type, ultrastructure, number, and distribution of sensilla on the mouthparts were described in detail. The differences in mouthpart morphology between mycophagous and predatory species were then discussed. This research can provide a reference for understanding the function of the mouthparts in ladybird feeding behavior and thus contribute to the development of precise insect behavior regulation and management strategies.

## 2. Materials and Methods

### 2.1. Insect Collection

Adult lady beetles, *I. chinensis* (Coleoptera: Coccinellidae), were collected from Kunming, Yunnan Province, China, on 7 December 2021. The adults were stored in 75% ethanol solution and then stored at 4 °C in a refrigerator before use.

### 2.2. Scanning Electron Microscopy

Ten female and ten male adults were randomly selected from the preserved specimens and then rinsed twice with 75% ethanol using an ultrasonic cleaner (SB-5200DTD, Scientz, Ningbo, China), with each rinse lasting for twenty seconds. The heads of the samples were then dissected from the bodies and gradually dehydrated in 80%, 85%, 90%, and 95% ethanol for twenty minutes at each concentration; this was followed by two twenty-minute periods in 99.9% ethanol. After this, each mouthpart was removed from the head with fine forceps under a stereomicroscope (Stemi 305, Zeiss, Suzhou, China). Mouthparts were then placed into a clean petri dish and stored in a 40 °C electric thermostatic drying oven (GZX-GF101-2-BS-II/H, Hengzi, Shanghai, China) for twelve hours until all samples were totally dry. The dried samples were mounted on aluminum stubs with double-sided copper sticky tape, and then they were coated with gold for one minute using a high-resolution sputter coater (ACE600, Leica, Vienna, Austria). A scanning electron microscope (SU8010, Hitachi, Tokyo, Japan) was used to observe the samples and take photos at 5 KV.

### 2.3. Image Processing and Data Analysis

Images were imported into Adobe Photoshop 2019 (Adobe Systems, San Jose, CA, USA) for processing and measurement. The lengths and basal diameters of the sensilla were determined using at least ten individual sensilla of the same type obtained from different samples. For the identification of various types of sensilla, the traditional morphological classification method was used according to the external morphology, length, and distribution [4,25]. After detailed analysis and comparison, we determined that there was no difference between male and female samples with respect to mouthpart morphology and sensilla; therefore, the images shown were mainly from the female samples.

## 3. Results

### 3.1. Gross Morphology of the Mouthparts

The head of the *I. chinensis* adult is nearly round and almost completely covered by the pronotum (Figure 1a,b). It bears a pair of compound eyes, a pair of antennae, and a mouthpart at the forefront (Figure 1c). The basic components of the mouthparts of *I. chinensis* are similar to those in other ladybeetles, i.e., a labrum, a pair of mandibles, a pair of maxillae, a labium, and a hypopharynx. The hypopharynx is a non-sclerotized structure and was not taken into account in this study. From the dorsal side of the head, only the whole labrum and parts of the maxillary and labial palpi can be seen (Figure 1c); on the ventral side, the labium and maxillae can be seen (Figure 1d).

### 3.2. Types and Morphology of the Sensilla on the Mouthparts

From the scanning electron micrographs, it is obvious that the mouthparts of *I. chinensis* bear more sensilla than other areas of the body. In all, there are eight kinds of sensilla on the mouthparts of *I. chinensis*; these can be divided into fourteen types, including three types of sensilla chaetica (Sch), three types of sensilla basiconica (Sb), two types of sensilla styloconica (Sty), two types of sensilla coeloconica (Sco), one type of sensilla campaniformia (Sca), one type of sensilla placodea (Sp), one type of sensilla digitiformia (Sd), and one type of Böhm bristle (Bb) (Table 1).

Sensilla chaetica I (Sch1) are peg-like and upright. Inserted into a circular convex socket, they gradually becomes thinner from the base to the top, and their tips are quite sharp. Their surfaces are longitudinally grooved with no pore (Figure 2a and Figure 3e,g). Their lengths range from 52.8 to 60.2 μm, and their basal diameters range from 2.2 to 3.6 μm. This type of sensilla is the most widely distributed and can be found on the surface of all mouthparts (Table 1).

Sensilla chaetica II (Sch2) are similar to Sch1; they stand upright or are slightly bent at the end (Figure 2b). They are much longer and thicker than Sch1 and are distributed on the labrum and maxillae (Table 1).

Sensilla chaetica III (Sch3) are similar to Sch1 and Sch2; however, they are the longest of this kind of sensilla (Figure 2c) with a maximum length of 193.0 μm and a basal diameter of 5.2 μm. They are often distributed on the outer surfaces of maxillae and the labium (Table 1).

Sensilla basiconica I (Sb1) are coniform with a blunt tip and inserted straight into a circular socket (Figure 2d). Their surfaces are smooth with no pores. They are often relatively short (lengths range from 2.3 to 12.2 μm) and thin (basal diameters range from 1.4 to 2.6 μm). They can be found on the mandible, maxillae, and labial palpi (Table 1).

Sensilla basiconica II (Sb2) are similar to Sb1 but their bases are always slenderer (Figure 2e). Their basal sockets are round and concave with a slightly raised edge. Their lengths range from 15.3 to 29.7 μm, and their basal diameters range from 1.6 to 2.6 μm. They can only be found on maxillary palpi (Table 1).

Sensilla basiconica III (Sb3) are cylindrical and slightly bent with a blunt tip (Figure 2f). They are longer than Sb1 and Sb2 and are often gathered on the epipharynx of the labrum and the galea of maxillae. Their lengths range from 68.2 to 105.8 μm, and their basal diameters range from 3.4 to 5.0 μm (Table 1).

Sensilla styloconica I (Sty1) are densely distributed at the end of maxillary palpi and labial palpi and have a conical-like appearance. Sty1 is relatively strong, with multiple longitudinal grooves on the surface and a hole at the top. The upper parts of these sensilla are always divided into micro-digitations, which are gathered or separated by the outer longitudinal grooves. Their sockets are convex, with some globular processes in surrounding areas (Figure 2g,h and Figure 3d). Their lengths range from 3.6 to 5.0 μm, and their basal diameters range from 1.3 to 1.7 μm (Table 1).

Sensilla styloconica II (Sty2) are cylindrical and are only distributed on the maxillary palpi (Table 1). Similar to Sty1, their surfaces are also grooved, but their ends are flat with round protrusions. There is an obviously raised socket around the lower part of the cylindrical sensilla (Figure 2h). Their lengths and basal diameters are similar to those of Sty1.

Sensilla coeloconica I (Sco1) have the form of a bulbiform protrusion situated in a round concave. Their basal diameters range from 1.5 to 2.4 μm (Table 1). They are only distributed on the epipharynx (Figure 3a).

Sensilla coeloconica II (Sco2) are coniform, with wrinkled surfaces and smooth protrusions in the surrounding area (Figure 3b). Their lengths range from 5.7 to 11.4 μm, and their basal diameters range from 4.2 to 6.0 μm. They can only be found at the bottom of the epipharynx (Table 1).

Sensilla campaniformia (Sca) have a bell-like form, like a little bump situated on the central part of a round concave with a convex periphery. A pore can be found on the top of the bump, and many tiny pores can be found on the concave part (Figure 3b,c). In general, they are flat with a basal diameter of about 4.4 to 6.6 μm and are distributed only on the epipharynx of the labrum (Table 1).

Sensilla placodea (Sp) are round with slightly concaved edges. Their surfaces are smooth, and their diameters range from 1.1 to 3.2 μm (Figure 3d). They are often distributed around the sensilla area on the top of the labial palp (Table 1).

Sensilla digitiformia (Sd) look like fingers which are flat, slightly narrow in the middle, and situated in the longitudinal gap (Figure 3e). Their lengths range from 31.5 to 39.9 μm, and their basal diameters are about 2 μm. These sensilla are only distributed on the dorsal surface at the ends of maxillary palpi (Table 1).

Böhm bristles (Bb) are coniform, straight, and strong, with a smooth surface and a sharp tip that is located in a slightly round sunken socket (Figure 3f). Their lengths range from 7.0 to 11.1 μm, and their basal diameters range from 1.1 to 1.8 μm. They only gather on the basal part of the lacinia (Table 1).

### 3.3. Glandular Structures on the Mouthparts

Perforated plates (Pp) are sieve-like, with many tiny pores of different sizes on the surface. These structures are characterized by a diversity of shapes, and they may be round, triangular, oval, or in the shape of a football (Figure 2d and Figure 3g). They are universally distributed and can be found on the surfaces of all mouthparts (Table 1).

Cuticular pores (Cp) are round and deeply concave and are widely distributed on the surfaces of all mouthparts (Figure 2d and Figure 3d,g,h). They are quite small, with diameters typically less than 0.7 μm (Table 1).

### 3.4. Labrum

The labrum is an oblong-shaped double lamellar structure that is attached to the front margin of the anteclypeus (Figure 4a). Its outer surface is sculptured and bears various kinds of sensilla and glandular structures, such as Sch1, Sch2, Pp, and Cp (Figure 4b). The inner surface of the labrum is called the epipharynx. The surface of the epipharynx is relatively smooth and bears fewer sensilla than the outer surface. Often, two clusters of Sb3 are located on both sides of the front margin (Figure 4c,d). Sca, Sco1, and Sco2 are present in the central rectangular region (Figure 4e,f) and are always surrounded by different types of cuticular processes (e.g., coniform processes (Figure 4g), palmate processes (Figure 4h), and scaly processes (Figure 4i)). The same types of process may exhibit different morphologies according to different positions; for instance, palmate processes may have a single branch, two branches, or several branches (Figure 4h).

### 3.5. Mandible

Mandibles are pairs of highly sclerotized falciform structures that are located below the labrum. Their dorsal surfaces are smooth with large amounts of Pp and Cp randomly distributed. Sb1 and Sch1 can be observed on the outer margins of each mandible (Figure 5g). The condyles on the bottoms of both sides are mainly used for adduction and abduction of the mandibles. The dorsal condyle (Dc) is oval and partially covered by palmate processes (Figure 5c), while the ventral condyle (Vc) is hemispherical with a smooth surface and a cluster of hairy protrusions on the other side of the bottom (Figure 5f). The molar consists of two teeth, but there are some small differences between the left and right mandibles. In the left mandible, the molar is triangular, and the ventral tooth (Vt) is slightly smaller than the dorsal tooth (Dt) (Figure 5k). In the right mandible, the dorsal tooth of the molar is approximately triangular, while the ventral tooth is blunt (Figure 5j). Prostheca possess spiny processes on the dorsal side and multi-layered slender bristles on the ventral side (Figure 5h,i). The incisor of the mandible consists of two teeth. On the ventral side of the incisor, one accessory tooth can be seen on the first tooth, and 12–16 accessory teeth can be found on the second tooth (Figure 5l).

### 3.6. Maxillae

Maxillae are segmented structures located on both sides of the labium; they are composed of the cardo, stipes, lacinia, galea, and maxillary palpi (Figure 6a,b). The cardo is semicircular, and the stipes are rectangular. Several sensilla and glandular structures are distributed on their surfaces, such as Sch1 and Cp. The galea is divided into a basal region and a distal region. The distal region is a spoon-like structure with a reticular surface on the ventral side and a smooth surface on the dorsal side, with large amounts of Sb3 and Sch1 distributed on the top (Figure 6c). The lacinia is located close below the galea; on its ventral side, rows of Sb1 and Pp are arranged (Figure 6d). The dorsal side of lacinia is quite smooth with a lot of Bb gathered at the basal part (Figure 6e,f).

The maxillary palpi is segmented into four, with a basal segment articulating with the palpifer. The last segment is the most developed and is approximately fan-shaped on the lateral view. Both sides of the surface are scaly, with large amounts of Sch1 and numerous Sd (Figure 6g). The top region of the last segment looks like the bottom of a boat. It has abundant sensilla and epidermal processes. It can be divided into two regions according to the type of sensilla. The first region is covered with Sty1 and Sty2, and the second region is covered with Sb2 and many spiny processes (Figure 6h,i).

### 3.7. Labium

The labium is a thinner structure located on the central part of the ventral side of the mouthparts. It is composed of a postmentum, prementum, ligula, and labial palpi (Figure 7a). The postmentum (Pomt) is relatively smooth with several Sch1 on its central part (Figure 7a). The prementum (Prmt) is narrow at both ends and wide in the middle, with Sch3 and Cp on it (Figure 7c). The ligula is an inverted triangle with Sch1, Sb1, and Cp on it (Figure 7c,d). A pair of labial palpi is situated on both sides of the ligula. The end of the labial palp is round, with about 30 Sty1 and several Sb1, Sp, and Cp (Figure 7b). The dorsal surfaces of the ligula and prementum are completely covered with epidermal processes (Figure 7e,f).

## 4. Discussion

### 4.1. Variation of Mouthpart Morphology between Different Diets

The morphology of the mouthparts varies greatly between insects. Such variation may even be witnessed in similar types of mouthpart. Taking chewing mouthparts as an example, there are great differences in mouthpart morphology, and in sensilla types, between different families such as Anobiidae [26], Endomychidae [27], Nitidulidae [28], and Curculionidae [29]. Of course, such morphological differences in mouthparts, as with morphological differences in other body organs, may be caused by differences in taxonomic status. There is no doubt that the closer the taxonomic relationship, the more similar the morphology. From the extensive research which has been conducted into Coleoptera, it is obvious that the mouthparts of *I. chinensis* have most in common with those of *Coccinella transversoguttata* [3], *Hippodamia variegata* [21], and *Coccinella transversalis* [13], which are all from the same family. Their basal compositions, as well as the locations and shapes of each component, are very similar. However, a careful comparison reveals many differences. Undoubtedly, because of the importance of the mouth in feeding, such morphological differences in the mouthparts are closely related to diet.

Among the Coccinellidae, the main morphological differences in mouthparts have been found on the mandible [2], which is the first part to contact food. Samways found that the morphology of mandibles in ladybirds can be used as a basis for distinguishing between mycophagous, carnivorous, and phytophagous species [2]. The main differences involve the numbers of incisor teeth, the size of the molar, and the development of prostheca.

Incisors are usually thought to be used for cutting food, and incisor morphology is the key feature used to distinguish the diet of ladybirds [2,3,13,21]. Previous studies have shown that phytophagous ladybirds, such as *Afidenta alia,* have three or four large, blunt teeth in their incisors, with many accessory teeth usually found on the inner margin [2]. These accessory teeth are used to scrape the surfaces of leaves to ingest the plant juices [2]. In most predatory ladybirds, the mandible incisors usually bear two teeth, and the ventral tooth is smaller than the dorsal one; however, ladybirds that feed on scales have only one main tooth in their incisor [2,3,21]. In the case of mycophagous ladybirds, we found that the incisor of *I. chinensis* bears two teeth, as in predatory species; however, there is one lateral tooth on the dorsal tooth and 12–16 accessory teeth on the ventral tooth; these accessory teeth can be used to comb fungal spores [2].

In the feeding process, food is ground by the extrusion of the molar teeth [30,31]. The shape of the molar teeth is directly related to grinding efficiency. In predatory ladybirds, the two molar teeth are larger, sharper, and obviously separated [3,21]. In phytophagous species, the molar region is rounded and has no obvious basal tooth [2]. In mycophagous ladybirds, the molar of *I. chinensis* is composed of two teeth, as in predatory species, but is much smaller. This may be because the fungal spores are small enough to be digested, leading to degeneration of the molar teeth.

The lobe-like prostheca is mainly used to collect and transport food from the incisor area to the molar area [6,30]. Among the ladybirds, the morphology of this structure does not vary greatly [2,3,21]. In phytophagous coccinellids, these developed setae help in trapping plant juices. Among the mycophagous species, *I. chinensis* has comb-like prostheca similar to those of *Tytthaspis 16-punctata*; these comb-like setae can collect spores from the fungal hyphae and isolate them during movement [2,32].

Morphological variations in mouthparts other than mandibles are also related to diet. It is well known that the labrum protects other mouthpart structures and serves to prevent the outflow of food with the help of the labium. In predatory ladybirds, the width of the front edge of the labrum is similar to that of the base [3,21]. In the present study, we found that the front edge of the labrum of *I. chinensis* is obviously wider than the basal part and expands significantly on both sides. Moreover, *I. chinensis* has a ligula which is markedly expanded compared with predatory ladybirds. The wider labrum and expanded ligula may help to better protect food and prevent the outflow of spores. Turning to the maxillary palp, we find, in contrast to the predatory ladybirds, that the maxillary palp of *I. chinensis* is quite developed, especially the final segment, which is axe-shaped and extremely expanded in the edge. They have no obvious bottom boundary, and the sensory field expands outwards in order to better sense changes in the external environment and find spores quickly. In addition, an expanded galea apex, dense setae on the galea and lacinia, and a broadly truncated ligula help mycophagous ladybirds to collect spores and conidia [20,33].

In conclusion, we may state that many parts of the mouthparts of *I. chinensis* have undergone specialized adaptations for the consumption of fungal food. This phenomenon is also widespread among Coleoptera insects; among these, the morphology of mouthparts reveals varying degrees of specialization to efficiently exploit diverse food sources and adapt to different habitats. However, a comprehensive understanding of the functions of mouthparts requires the integration of functional information with morphological observations and behavioral studies. Relying solely on mouthpart morphology is insufficient for determining mouthpart functionality. Therefore, further investigations encompassing detailed morphological analysis, behavioral examinations, and analysis of intestinal contents are essential to elucidate the intricate relationship between insect mouthparts, feeding preferences, environmental factors, and sensory functions.

### 4.2. The Putative Function of the Sensilla

The sensilla on the mouthparts, especially on the maxillary and labial palpi, play an important role in food recognition, both in the process of finding food and feeding [3,21,34,35]. The sensilla types on these two sensory fields of *I. chinensis* are consistent with those in other predatory ladybirds, such as *C. transversoguttata*, *H. variegate*, *Coccinella septempunctata*, and *C. transversalis*, but notable differences can also be observed. First, two main types of sensilla (Sty1 and Sty2) can be found on the maxillary palpi of ladybirds. Only Sty1 can be found on the labial palpi, which is consistent among those species, but the quantity of sensilla is not the same. There is a greater abundance of Sty1 (twenty-eight) on the top of the labial palpi of *I. chinensis*; in contrast, only fifteen Sty1 were found in *C. transversoguttata* [3] and thirteen in *H. variegate* [21]. It has been suggested that Sty1 serve gustatory and mechanical functions and that Sty2 may serve as olfactory sensilla [3,4,13,21,36]. Second, no sensilla placodea are present at the tip of the maxillary palp, and only one sensilla placodea can be found on the top of labial palp; in contrast, in predatory species, numerous sensilla placodea are present at the tip of the maxillary palpi, and their labial palpi are always circled with sensilla placodea [3,21]. It has been suggested that sensilla placodea play a crucial role in registering cuticular stress exerted on palpal apices during prey capture and feeding [3,37]. Predatory ladybirds exhibit increased sensilla placodea to rapidly respond to external pressures and mitigate the effects of feeding objects due to their robust locomotion.

Sensilla digitiformia are neatly arranged on the dorsal side of the terminal segment of the maxillary palpi. This has never previously between reported in ladybirds, but it has been seen in other beetles such as *Xylotrechus grayii* [38] and *Phoracantha recurva* [16]. These sensilla may function as thermo-, hygro-, or CO_2_-receptors [39,40,41]. Additionally, electrophysiological tests have demonstrated that sensilla digitiformia also function as tactile mechanoreceptors for detecting contact and vibratory stimuli [42]. It has been suggested that similar sensilla in *Nitidula carnaria* are both mechanoreceptors and chemoreceptors involved in hygroreception [43]. In *Omosita colon*, these sensilla are referred to as sensilla placodea, and their olfactory function has been described elsewhere [14].

There are two kinds of glandular structures on the mouthparts of *I. chinensis*. Cuticular pores are commonly observed in ladybirds and are widely distributed in various mouthparts. Sevarika observed the same pores on the antennae of *Harmonia axyridis* using a transmission electron microscope, confirming that these pores are terminal apparatus at the level of secretory cells [44]. In the present study, some strips of secretion were observed coming out of these pores, demonstrating the same function of Cp reported in a previous study [44]. Perforated plates are widely distributed on all mouthparts, especially on the dorsal sides of the mandibles. Similar structures have been observed in *H. axyridis*; further examination via transmission electron microscopy revealed a sub-cuticular chamber without specialized evacuation ducts [44]. Glandular cells are excreted onto the cuticle surface through these pores [45]. The same structures have also been found in many other insects, including *Paussus favieri* [46], *Drilus mauritanicus* [47], *Graphoderus occidentalis* [48], and some species of Erotylinae [49].

The other sensilla are prevalent in ladybirds and various other insects, and their functional significance has been firmly established. Sensilla basiconica are typically involved in taste perception and food detection [40,50]. In the present study, Böhm bristles were found on the lacinia, as they are similarly found in *Omosita colon* [14]. Researchers have demonstrated that these Böhm bristles function as gravity-receptor sensilla which respond to external stimuli when insects experience gravitational forces [51]. As is the case in other ladybirds, the absence of pores on the surface of sensilla chaetica suggests a mechanoreceptor role [52]. Sensilla coeloconica is considered to function in chemical perception and sense changes in temperature and humidity [53,54,55]. Sensilla campaniformia is associated with proprioception. When in contact with food, they respond to the tensions and strains of the cuticle and detect any deformation of the cuticle [7,16,56].

In the classification and identification of sensilla, we often found that sensilla with a similar morphology in different species were identified as different types, while those sensilla with diverse morphologies may be assigned the same name by various researchers. Moreover, the shape, length, and distribution of sensilla do not always directly correlate with their function [57,58]. Consequently, the classification of sensilla and the confirmation of their function requires not only external morphology but also more research on the structural characteristics of functional relevance [25]. Therefore, it is crucial to establish a more detailed and unified standard for the classification of insect sensilla types and the exploration of their function.

## Figures and Tables

**Figure 1 insects-15-00046-f001:**
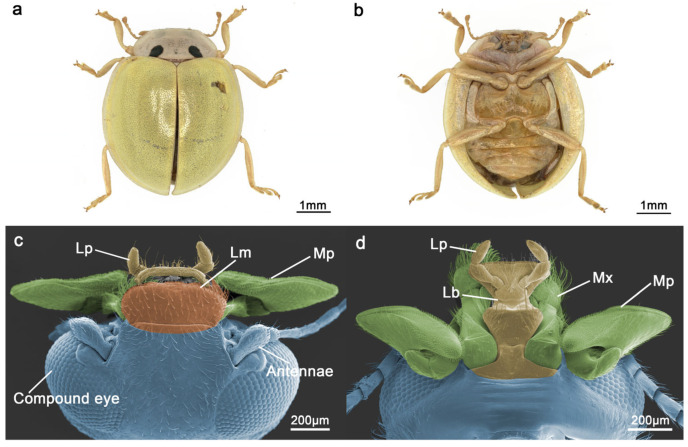
The overall morphology of *Illeis chinensis* adults and the mouthparts. (**a**) Dorsal view of the adult. (**b**) Ventral view of the adult. (**c**) Dorsal view of the head showing the position and morphology of the compound eyes, antennae, labrum (Lm), labial palpi (Lp), and maxillary palpi (Mp). (**d**) Ventral view of the mouthparts showing the position and morphology of maxillae (Mx), maxillary palpi (Mp), labium (Lb), and labial palpi (Lp).

**Figure 2 insects-15-00046-f002:**
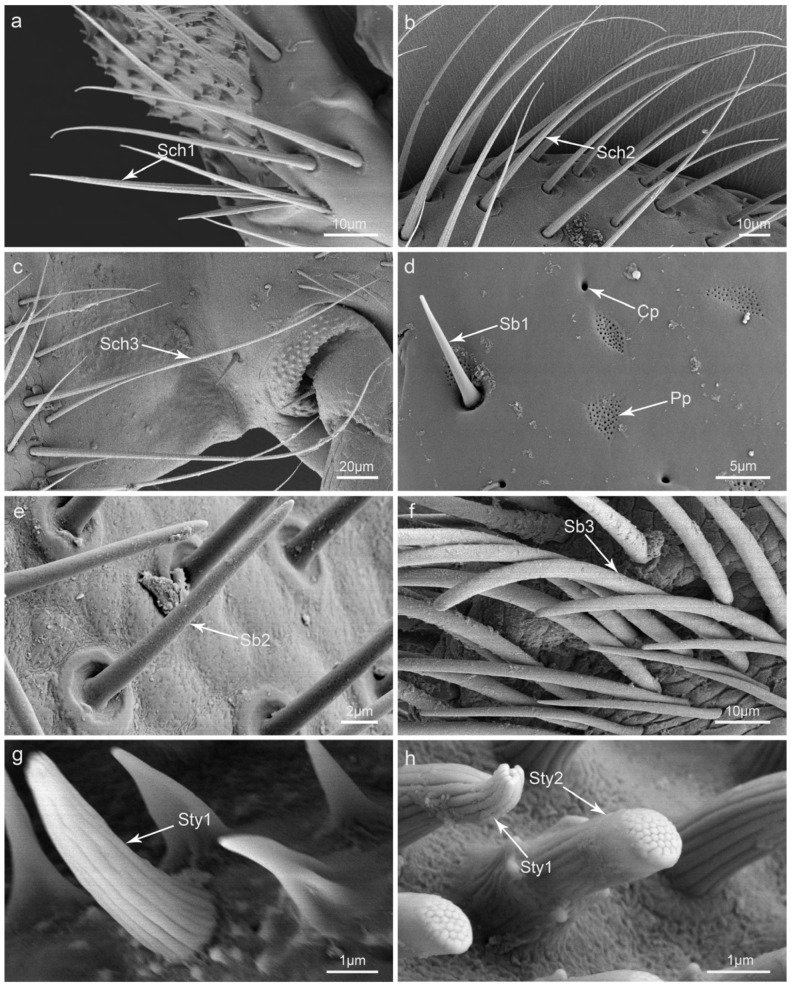
Scanning electron micrographs of different kinds of sensilla and glandular structures on the mouthparts of *Illeis chinensis*: (**a**) sensilla chaetica I (Sch1); (**b**) sensilla chaetica II (Sch2); (**c**) sensilla chaetica III (Sch3); (**d**) sensilla basiconica I (Sb1), cuticular pores (Cp), and perforated plates (Pp); (**e**) sensilla basiconica II (Sb2); (**f**) sensilla basiconica III (Sb3); (**g**) sensilla styloconica I (Sty1); (**h**) sensilla styloconica I (Sty1) and sensilla styloconica II (Sty2).

**Figure 3 insects-15-00046-f003:**
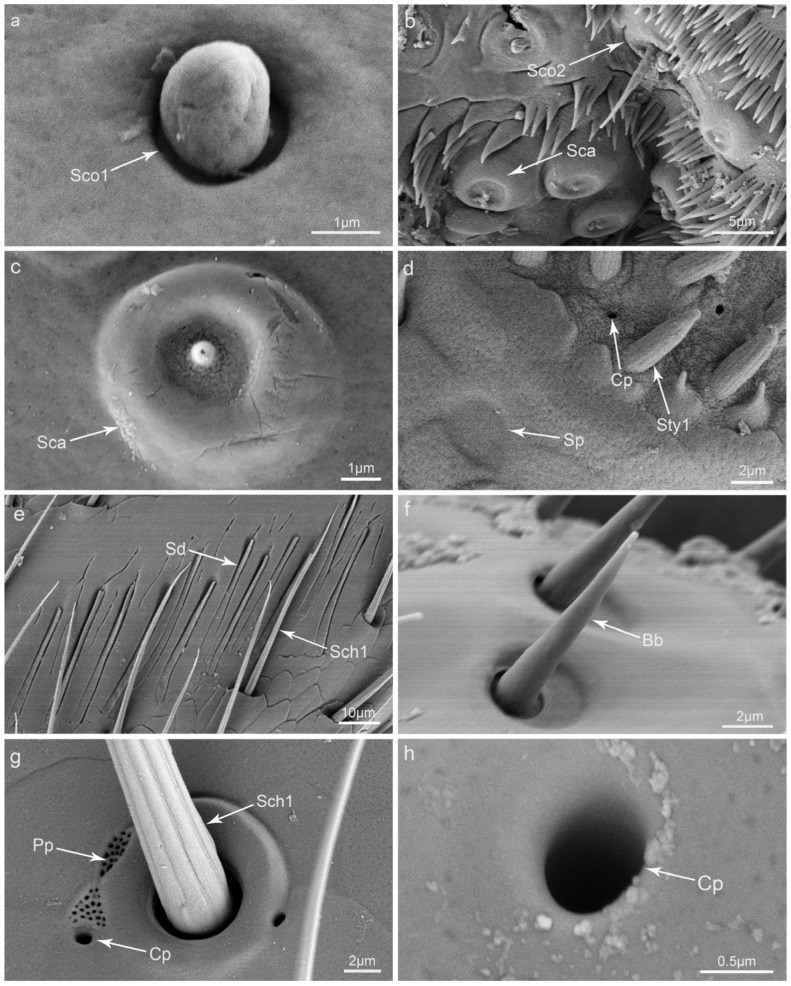
Scanning electron micrographs of the sensilla and glandular structures on the mouthparts of *Illeis chinensis*: (**a**) sensilla coeloconica I (Sco1); (**b**) sensilla coeloconica II (Sco2) and sensilla campaniformia (Sca); (**c**) sensilla campaniformia (Sca); (**d**) sensilla styloconica I (Sty1), sensilla placodea (Sp), and cuticular pores (Cp); (**e**) sensilla digitiformia (Sd) and sensilla chaetica I (Sch1); (**f**) Böhm bristle (Bb); (**g**) perforated plates (Pp), sensilla chaetica I (Sch1), and cuticular pores (Cp); (**h**) cuticular pores (Cp).

**Figure 4 insects-15-00046-f004:**
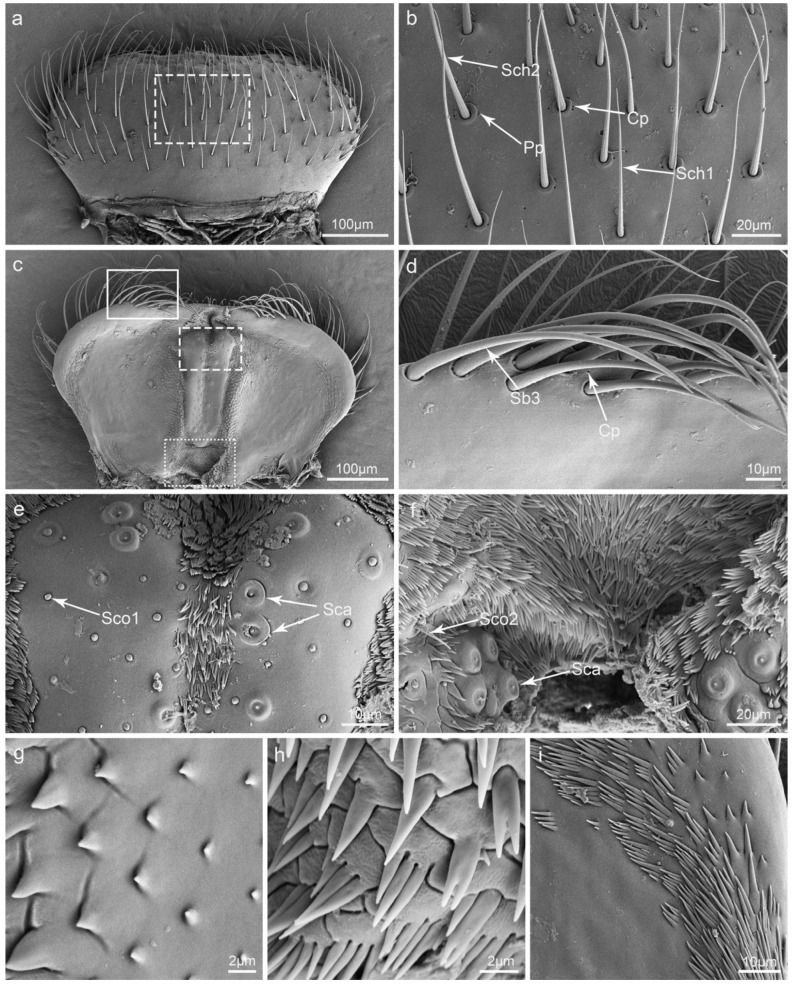
Scanning electron micrographs of the labrum of *Illeis chinensis*. (**a**) Dorsal view of the labrum; (**b**) enlarged view of the dashed box in (**a**) showing sensilla chaetica I (Sch1), sensilla chaetica II (Sch2), perforated plates (Pp), and cuticular pores (Cp); (**c**) ventral view of the labrum; (**d**) enlarged view of the solid box in (**c**) showing sensilla basiconica III (Sb3) and cuticular pores (Cp); (**e**) enlarged view of the dashed box in (**c**) showing sensilla campaniformia (Sca) and sensilla coeloconica I (Sco1); (**f**) enlarged view of the dotted box in (**c**) showing the sensilla campaniformia (Sca) and sensilla coeloconica II (Sco2); (**g**) coniform processes; (**h**) palmate processes; (**i**) scaly processes.

**Figure 5 insects-15-00046-f005:**
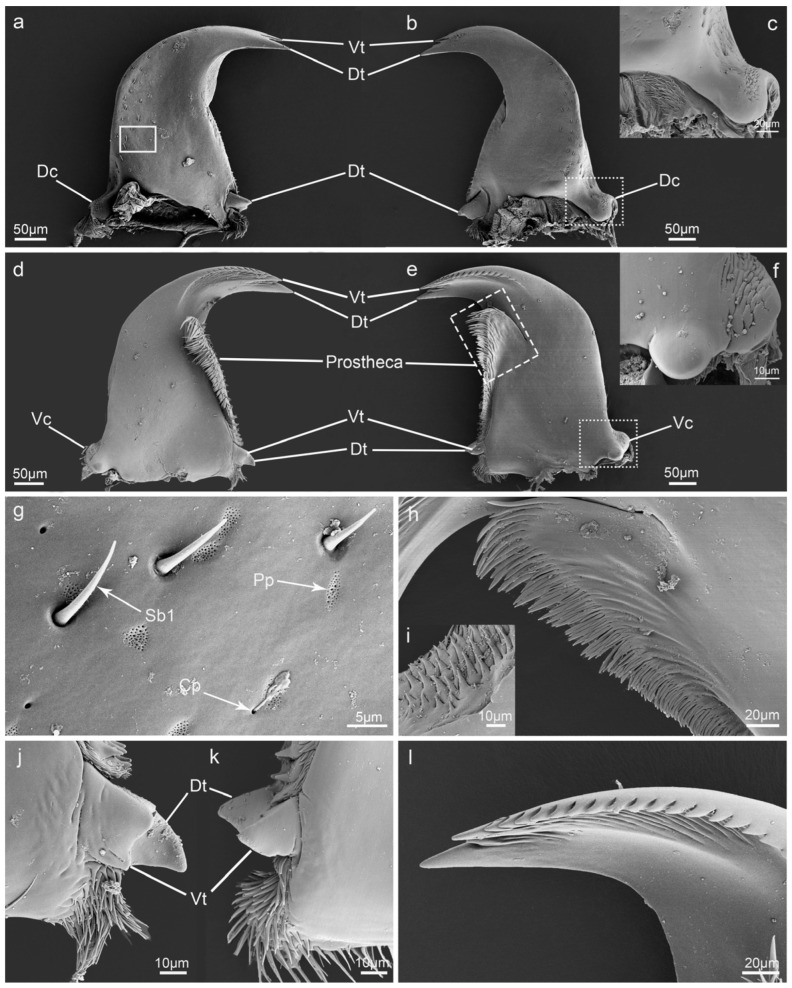
Scanning electron micrographs of the mandible of *Illeis chinensis*. (**a**) Dorsal view of the left mandible; (**b**) dorsal view of the right mandible; (**c**) enlarged view of the point line box in (**b**) showing dorsal condyle (Dc); (**d**) ventral view of the right mandible; (**e**) ventral view of the left mandible showing ventral teeth (Vt), dorsal teeth (Dt), prostheca, and ventral condyles (Vc); (**f**) enlarged view of the point line box in (**e**) showing ventral condyles (Vc); (**g**) enlarged view of the solid box in (**a**) showing the sensilla basiconica I (Sb1), perforated plates (Pp), and cuticular pores (Cp); (**h**) enlarged view of the dotted line box in (**e**) showing the ventral view of prostheca; (**i**) spiny processes on the dorsal side of prostheca; (**j**) ventral view of the molar teeth of right mandible; (**k**) ventral view of the molar teeth of left mandible; (**l**) ventral view of incisor.

**Figure 6 insects-15-00046-f006:**
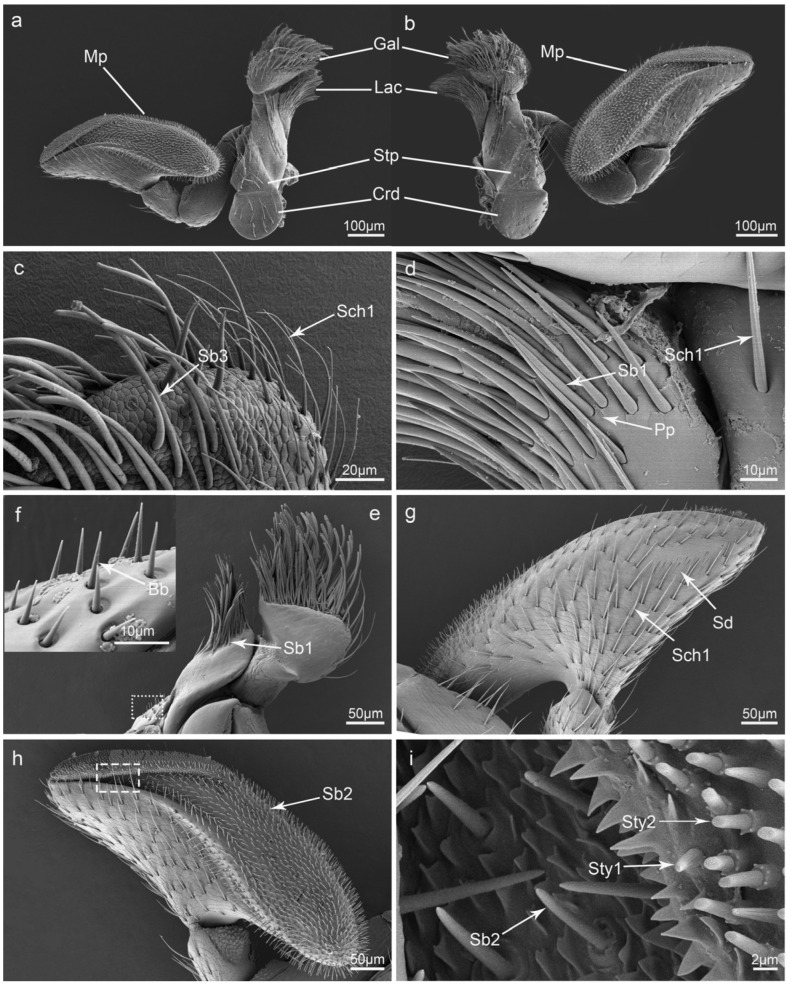
Scanning electron micrographs of the maxillae of *Illeis chinensis*. (**a**) Ventral view of the right maxillae showing maxillary palp (Mp), galea (Gal), lacinia (Lac), stipes (Stp), and cardo (Crd); (**b**) ventral view of the left maxillae; (**c**) ventral view of the tip of galea showing sensilla chaetica I (Sch1) and sensilla basiconica III (Sb3); (**d**) ventral view of the lacinia showing sensilla chaetica I (Sch1), sensilla basiconica I (Sb1), and perforated plates (Pp); (**e**) dorsal view of the galea and lacinia showing sensilla basiconica I (Sb1); (**f**) enlarged view of the point line box in (**e**) showing the Böhm bristle (Bb); (**g**) dorsal view of the right maxillary palp showing sensilla chaetica I (Sch1) and sensilla digitiformia (Sd); (**h**) ventral view of the maxillary palp showing sensilla basiconica II (Sb2); (**i**) enlarged view of the dashed box in (**h**) showing the sensilla styloconica I (Sty1), sensilla styloconica II (Sty2), and sensilla basiconica II (Sb2).

**Figure 7 insects-15-00046-f007:**
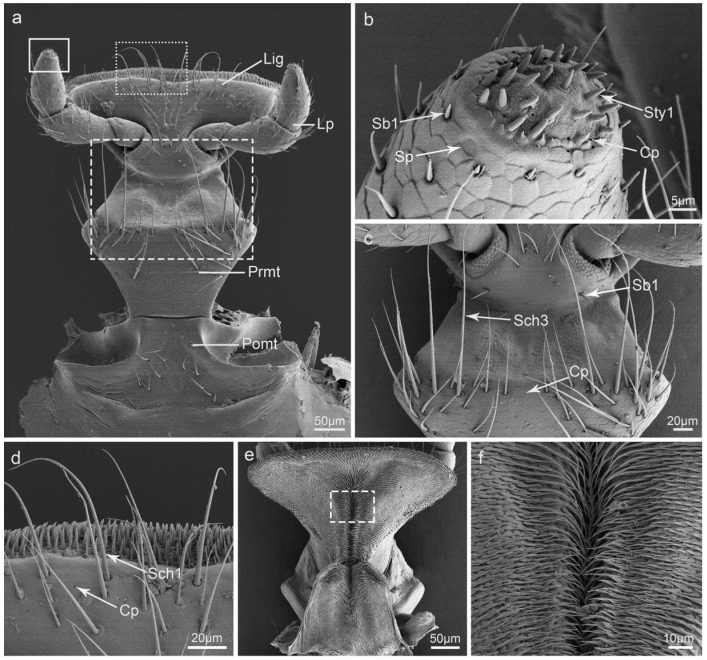
Scanning electron micrographs of the labium of *Illeis chinensis*. (**a**) Ventral view of the labium showing ligula (Lig), labial palp (Lp), prementum (Prmt), and postmentum (Pomt); (**b**) enlarged view of the solid box in (**a**) showing the sensilla styloconica I (Sty1), sensilla basiconica I (Sb1), sensilla placodea (Sp), and cuticular pores (Cp); (**c**) enlarged view of the dashed box in (**a**) showing the sensilla chaetica III (Sch3), sensilla basiconica I (Sb1), and cuticular pores (Cp); (**d**) enlarged view of the point line box in (**a**) showing sensilla chaetica I (Sch1) and cuticular pores (Cp); (**e**) Dorsal view of the labium; (**f**) enlarged view of the dashed box in (**e**) showing epidermal processes.

**Table 1 insects-15-00046-t001:** The morphological characters of the sensilla and glandular structures on the mouthparts of *Illeis chinensis*.

Type		Shape	Socket	Surface	Pore	Length (μm)	Diameter (μm)	Distribution
Sensilla	Sch1	Peg	Convex	Grooved	No	52.8–60.2	2.2–3.6	Lm, Md, Mx, Lb
	Sch2	Hair, peg	Convex	Grooved	No	95.2–110.0	3.4–5.6	Lm, Mx
	Sch3	Hair, peg	Convex	Grooved	No	91.4–193.0	3.1–5.2	Mx, Lb
	Sb1	Coniform	Concave	Smooth	No	2.3–12.2	1.4–2.6	Md, Mx, Lp
	Sb2	Coniform	Concave	Smooth	No	15.3–29.7	1.6–2.6	Mp
	Sb3	Hair, cylindrical	Concave	Smooth	No	68.2–105.8	3.4–5.0	Epi, Gal
	Sty1	Conical	Convex	Grooved	Apical pore	3.6–5.0	1.3–1.7	Mp, Lp
	Sty2	Cylindrical	Convex	Grooved	Apical pore	2.6–4.2	1.5–1.8	Mp
	Sco1	Coniform	Convex	Smooth	No	–	1.5–2.4	Epi
	Sco2	Coniform	Convex	Rugose	No	5.7–11.4	4.2–6.0	Epi
	Sca	Round	Convex	Papilliform	Multiporous	–	4.4–6.6	Epi
	Sp	Round	Concave	Smooth	No	–	1.1–3.2	Lp
	Sd	Finger	Concave	Smooth	No	31.5–39.9	1.8–2.3	Mp
	Bb	Conical	Concave	Smooth	No	7.0–11.1	1.1–1.8	Lac
Glandular structures	Pp	Irregular	Concave	No	Multiporous	-	1.6–5.1	Lm, Md, Mx, Lb
	Cp	Hole	Concave	-	Uniporous	-	0.4–0.7	Lm, Md, Mx, Lb

Sch—sensilla chaetica; Sb—sensilla basiconica; Sty—sensilla styloconica; Sco—sensilla coeloconica; Sca—sensilla campaniformia; Sp—sensilla placodea; Sd—sensilla digitiformia; Bb—Böhm bristle; Pp—perforated plates; Cp—cuticular pores; Lm—labrum; Md—mandible; Mx—maxillae; Lb—labium; Mp—maxillary palp; Lp—labial palp; Epi—epipharynx; Lac—lacinia; Gal—galea.

## Data Availability

The data presented in this study are available in the article.

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
