# Peer review of "The Microscopic Morphology of Mouthparts and Their Sensilla in the Mycophagous Ladybeetle Illeis chinensis (Coleoptera: Coccinellidae)"

_insects, 2024, doi:10.3390/insects15010046_

Round 1
Reviewer 1 Report
Comments and Suggestions for Authors
This manuscript is interesting and well written. The micrographs are very beautiful.
There are several errors in identifyng the sensilla. (see report).

I am not qualified.
Author Response
Reviewer #1:
The paper concerns the morphology of mouthparts of a mycophagous ladybeetle Illeis chinensis. This research can provide a reference for understanding the function of the mouthparts in ladybeetle feeding behavior.
1. There are several errors in identifying the sensilla:
(1) Sensilla placodea 1 (Sp1) are not sensilla but glandular structures named « perforated plates » or « pore plates » which exist for example in Staphylinidae: antennae of Aleochara (Skilbeck & Anderson 1994); Carabidae : antennae of Paussus favieri (Di Giulo et al. 2009); Erotylidae: antennae of Erotylinae (Drilling et al. 2013); Drilini (Elateridae Agrypninae Drilini): male antennae of Drilus mauritanicus (Faucheux & Kundrata 2015), Malacogaster nigripes (Faucheux 2016), and other male Drilini (Faucheux & Kundrata (2017); labrum-epipharynx and maxillary palps of male Drilus mauritanicus (Faucheux 2017); and Dytiscidae: labial palp of Graphoderus occidentalis (Leung & Zacharuk 1986). Indeed, sensilla placodea on antennae of Hymenoptera are all similar in the same species and not « triangular, oval, or round » and their pores are not of different sizes. Curiously, in Discussion, the presumed function remains correct.
Response: I greatly appreciate your valuable suggestions, especially in identifying the sensilla. Sp1 has been revised as perforated plates as suggested. We have thoroughly read the literature you recommended and incorporated them into our article. We apologize unreservedly for some errors in sensilla identification due to a lack of transmission electron microscopy observation. The function of sensilla were derived from extensive references. In subsequent stages, we will pay close attention to the internal structures of various sensilla, in order to accurately ascertain their functions. We would like to express our sincere appreciation once again for your insightful suggestions. The description of perforated plates has been rewritten in line 207-210, and the discussion has been described in detail in line 409-415.
(2) Cuticular pores (Cp) are not sensilla but openings of epidermal glands. Curiously, the function of Cp is well established.
Response: Thank you for your suggestion. Cuticular pores has been classified as glandular structure and descriped separately in line 211-213. The discussion about their function has been described in detail in line 403-409.
(3) Consequently, delete « Spl » and « Cp » in 3.2 Sensilla and Table 1; and write a new paragraph 3.3 Glandular structures on mouthparts (see also in the Abstract ,…).
Response: Thank you for your suggestion. This section has been rewritten as suggested. Sp1 has been revised as perforated plates (Pp) as suggested. « Pp » and « Cp » in Table 1 are listed as Glandular structures in line 129-132.
(4) Sensilla placodea (Sp2) appear correct but a magnification would be welcome. SP2 becomes Sp1.
Response: Thank you for your suggestion. Sp2 has been renamed as Sp as suggested. The magnification of Sp can be seen in Figure 3d. In the process of observation under SEM, we have magnified Sp2 to see more details, but no pore or other structure has been found. The surface of Sp is similar to the cuticle around. Finally, we chose this picture because all three types of sensilla on it are quite clear.
(5) Styloconica 2 (Sty2) are indeed sensilla styloconica because they possess a high stylus (= very high base) but Styloconica 1 (Sty1) do not have a stylus and are well known grooved peg sensilla.
Response: Thank you for your suggestion. These two typs of sensilla are universal in ladabirds, and we have found the seme sensilla in many other species such as Coccinella transversoguttata (Hao et al. 2019) and Hippodamia variegata (Hao et al. 2020). Their morphology and distribution are quite stable in different species. In other species, we have clearly observed that Sty1 possess high stylus, which is not as high as Sty2. In addition, we can find multi-finger structure on the top of Sty1 in this study, which is obviously different from grooved peg sensilla. Therefore, we have retained the names of Sty1 and Sty2.Hao, Y.N.; Sun, Y.X.; Liu, C.Z. Functional morphology of the mouthparts of lady beetle Coccinella transversoguttata (Coccinellidae, Coleoptera), with reference to their feeding mechanism. J. Morphol. 2019, 280, 701–711.Hao, Y.N.; Sun, Y.X.; Liu, C.Z. Functional morphology of the mouthparts of lady beetle Hippodamia variegata (Coleoptera: Coccinellidae), with reference to their feeding mechanism. Zoomorphology 2020, 139, 199–212.
(6) Sensilla campaniformia 1, 2 are not sensilla campaniformia but for Sca1: uniporous dome- shaped sensillum basiconicum and for Sca2: short sensillum chaeticum. Sensilla campaniformia of insects are well defined (Mclver SB 1985): a central dome-shaped sensory part surrounded by a cuticular ring; and their function is always a proprioceptive function. There are many errors in the current scientific literature. Figure 3C of Illeis chinensis is noteworthy!
Response: Thank you for your suggestion. According to the description of Sca in (Mclver SB 1985), We consider that Sca1 fit this description and should be named as Sca. This type of sensilla is quite common and has been found in the same place of mouthpart in other ladybirds. Sca2 has been renamed as Sco2 because the central part of these sensilla are smooth without groooves on the surface, while sensilla chaetica bare obvious longitudinal groove on their surface. The socket of these sensilla are round and slightly concave just like Sco1. In addition, according to our observation of ladybirds, no other type of sensilla was found on epipharynx other than sensilla campaniformia and sensilla coeloconica. Therefore, we defined Sca2 into Sco2 in line 179-182.
2. Remarks in the text
(1) 23, mouthparts and all in the text
Response: Thank you for your suggestion. All the “ mouthparts ” in the text has been checked and corrected as suggested.
(2) 25-26, to modify (see report)
Response: Thank you for your suggestion. This sentence has been modify as suggested in line 25-27.
(3) 45, sensilla is plural of sensillum
Response: Thank you for your suggestion. All the “ sensilla ” in the text has been checked and corrected as suggested.
(4) 46, they exhibit
Response: Thank you for your suggestion. This sentence has been modified as suggested in line 51.
(5) 54, Cerambycidae [14,add:Faucheux 2013,Liu&Tong 2023
Response: Thank you for your suggestion. This literature has been added as suggested in line 59-60.
(6) 64, feeding organs, mouthparts
Response: Thank you for your suggestion. This sentence has been modified as suggested in line 69.
(7) 65, crucial
Response: Thank you for your suggestion. This sentence has been modified as suggested in line 70.
(8) 101, 103, mouthparts
Response: Thank you for your suggestion. This sentence has been modified as suggested in line 106, 109.
(9) 118, fifteen
Response: Thank you for your suggestion. This sentence has been modified as suggested in line 124.
(10) 121, Böhm sensilla is better than Böhm bristles
Response: Thank you for your suggestion. This type of sensilla is common in many insects, and is known as Böhm bristles, such as Cao et al. 2016 and Li et al. 2021.Cao, Y.K.; Huang, M. A SEM study of the antenna and mouthparts of Omosita colon (Linnaeus) (Coleoptera: Nitidulidae), Microsc. Res. Tech. 2016, 79, 1152-1164.Li, Q.H.; Chen, L.Y.; Liu, M.K.; Wang, W.K.; Sabatelli, S.; Di Giulio, A.; Audisio, P. Scanning Electron Microscope Study of Antennae and Mouthparts in the Pollen-Beetle Meligethes (Odonthogethes) chinensis (Coleoptera: Nitidulidae: Meligethinae). Insects 2021, 12, 659.
(11) 122, cuticular pores are not sensilla but openings of glands
Response: Thank you for your suggestion. This sentence has been deleted as suggested.
(12) 159-160, protrusions
Response: Thank you for your suggestion. This sentence has been modified as suggested in line 167-168.
(13) 190. Böhm sensilla are ……
Response: Thank you for your suggestion. This type of sensilla is common in many insects, and is known as Böhm bristles, such as Cao et al. 2016 and Li et al. 2021.Cao, Y.K.; Huang, M. A SEM study of the antenna and mouthparts of Omosita colon (Linnaeus) (Coleoptera: Nitidulidae), Microsc. Res. Tech. 2016, 79, 1152-1164.Li, Q.H.; Chen, L.Y.; Liu, M.K.; Wang, W.K.; Sabatelli, S.; Di Giulio, A.; Audisio, P. Scanning Electron Microscope Study of Antennae and Mouthparts in the Pollen-Beetle Meligethes (Odonthogethes) chinensis (Coleoptera: Nitidulidae: Meligethinae). Insects 2021, 12, 659.
(14) 194, Cuticular pores are …
Response: Thank you for your suggestion. This sentence has been modified as suggested in line 211.
(15) 281, palps on ligula
Response: Thank you for your suggestion. This sentence has been modified as suggested in line 293
(16) 361, The sensilla on mouthparts
Response: Thank you for your suggestion. This sentence has been modified as suggested in line 375.
(17) 363, The sensilla types …...are consistant …
Response: Thank you for your suggestion. This sentence has been modified as suggested in line 377.
(18) 371-376, « sensilla placodea Spl « of Illeis are glandular pore plates
Response: Thank you for your suggestion. This section has been rewritten as suggested. Sp1 has been revised as perforated plates (Pp) as suggested. The description of perforated plates has been rewritten in line 207-210, and the discussion has been described in detail in line 409-415.
(19) 381, [35] and Phoracantha recurva [Faucheux 2013]
Response: Thank you for your suggestion. This literature has been added as suggested in line 396.
(20) 384, carnaria
Response: Thank you for your suggestion. This sentence has been modified as suggested in line 399.
(21) 388, Cuticular pores are
Response: Thank you for your suggestion. This sentence has been modified as suggested in line 403-404.
(22) 389-391, « classified as one type sensilla…axyrids [41]. Sevarika et al (2021) are wrong. Delete this sentence.
Response: Thank you for your suggestion. This sentence has been deleted and this part has been rewritten as suggested in line 403-409.
(23) 393-394, However……studied: delete. Cuticular pores hazve not any sensory function.
Response: Thank you for your suggestion. This sentence has been deleted as suggested.
(24) 397-400: to delete
Response: Thank you for your suggestion. This sentence has been deleted as suggested.
(25) 403-405, it is true for sensilla chaetica but sensilla coeloconica of the epipharynx probably are not mechanoreceptive but rather chemoreceptive (terminal pore or wall pores are not always visible in SEM).
Response: Thank you for your suggestion. This section has been rewritten as suggested in detail in line 422-424.
(26) 465-570: many mistakes and omissions.
Response: Thank you for your suggestion. All the references have been checked and corrected.
(27) 470, write: Abd El Ghany, N.M.; Abd El Aziz, E. … … … … …
Response: Thank you for your suggestion. This sentence has been modified as suggested in line 505.
References to add:
- Di Giulio, A.; Rossi Stacconi, M.V.; Romani, R. Fine structure of the antennal glands of the ant nest beetle Paussus favieri (Coleoptera, Carabidae, Paussini). Struct. Dev. 2009, 38, 293-302.
- Drilling, K.; Dettner, K.; Klass, K.D. The distribution and evolution of exocrine compounds glands in Erotylinae (Insecta: Coleoptera: Erotylidae). Soc. Ent. Fr. (N.S.). 2013, 1, 36-52.
- Faucheux, M.J. Mouthpart sensilla of the adult yellow longicorn beetle Phoracantha recurva Newman, 1840 (Coleoptera, Cerambycidae, Cerambycinae). Inst. Scientif., Rabat, Sect. Sci. Vie. 2013, 35, 83–94.
- Faucheux, M.J. Single pores and perforated plates of antennal aphrodisiac glands in winged male imagos Drilus mauritanicus Lucas 1849 and Malacogaster nigripes Schaufuss 1867 (Coleoptera: Elateridac: Agrypninae: Drilini). Soc. Sci. Nat. Ouest Fr. (N.S.) 2016, 38(hors-série 1), 165-177.
- Faucheux, M.J. Mouthpart sensilla of the male imago of Drilus mauritanicus Lucas 1849 (Coleoptera: Elateridae: Agrypninae: Drilini). Soc. Sci. Nat. Ouest Fr. (N.S.) 2017, 39 (hors-série 1): 1-21.
- Faucheux, M.J; Kundrata, R. Perforated plates corresponding to integumental glands on the antennae of adult male Drilus mauritanicus Lucas 1849 (Coleoptera: Elateridae: Agrypninae: Drilini). Soc. Ent. Fr. (N.S.). 2015, 1, 1-3.
- Faucheux, M.J.; Kundrata, R. Comparative antennal morphology of male Drilini with special reference to the sensilla (Coleoptera: Elateridae: Agrypninae). Anz. 2017, 266: 105-119.
- Leung, E.S.; Zacharuk, R.Y. Fine structure of integumental glands in the labial palp of adult Graphoderus occidentalis Horn (Coleoptera: Dytiscidae). J. Zool. 1986, 64, 2788-2800.
- Liu, C.T.; Tong, X. Functional morphology of the mouthparts of longhorn beetle adult Psacothea hilaris (Coleoptera:Cerambycidae) and sensilla comparisons betwen the sexes. Struct. Dev. 2023, 77, 101312.
- Mclver, S.B. Mechanoreception, 71-132. In G.A Kerkut & L.I. Gilbert(eds.), Comprehensive Insect Physiology, Biochemistry and Pharmacology, vol.6, Pergamon Press, Oxford, 1985.
- Skilbeck, C.A.; Anderson, M. The fine structure of glandular units on the antennae of two species of the parasitoid, Aleochara (Coleoptera: Staphylinidae). J. Ins. Morph. Embryol. 1994, 23, 319-328.
Response: Thank you for listing the literature, Most of the reference have been carefully read and added into the text. However, the fourth and fifth reference can not be found maybe because of different language.
Reviewer 2 Report
Comments and Suggestions for Authors
1) Introduction – can you add some literature about general insect morphology? Not every reader knows insect morphology.
2) line 90: “Scanning electron microscopy - cope (SU8010, Hitachi, Japan) was used to observe and take photos at 5KV – there should be the equipment, not the technology.
3) Results – line 107-108 – there is a reference to Fig. 1c and 1d, but not for 1a and 1b. These figures are also described below (line 111) as Adult with capital letter, which is not needed, lowercase letters are sufficient.
4) line 115: Types and morphology would be more precise.
5) line 116: “According to the scanning electron microscopy (…)” SEM cannot suggest anything being machine – I suggest to change for: “After analyzing under SEM it is (…)”.
6) line 120: shortcut Sty (for s. styloconica)– the abbreviation SS is most common in the literature and I suggest to use it in this article.
7) line 156: “There sockets are convex with some globular process around (Figure. 2g, h and 3b)”. The convex socket is hardly visible, can you provide one bette3r shot for this element?
8) line 172: can you provide better figure for Sp2? In the Fig. 3b there is some trace/imprint of poor quality, easy to confuse with anything else.
9) line 232: lack of a verb: Prostheca dorsally (has/possess?) spiny processes, ventrally multi-layered slender bristles, ventrally 232 from the molar to the abdomen
10) line 254: too long sentence: “The lacinia bears setae and spines 253 on its margin and at its apex. and is closely located below the galea, and a row of Sb1 are 254 arranged at the edge of lacinia, dorsal smooth, ventral distribution of Sch1, Sb1, Sp1, Bb, 255 Cp (Figure. 6d, e, f)”. What about the dot after the apex?
11) line 274: The Cp is not marked on the photo 6h, i - is it a mistake?
12) line 280: ligula should be with a capital letter, and also sensilla (Sch1, Sb1, Cp) are not shown – can you provide such figure?
13) line 282: “The ventral surface of labial palp is completely covered by epidermal processes, including water droplet processes, conical processes, and hairy processes (Figure. 7d, e)” – you wrote about the ventral surface of Lp, while Fig. 7d, e are about Lig and Prmt and also there is lack of any described processes marked.
14) lines 288-289 – there is a lack of Sp2 – it is in the Fig. 7b.
15) line 328: there is some information about prostheca – can you move general info about it to the beginning of the article?
16) line 384: Nitidula carnaria – the specific name should be with lowercase letter.
Author Response
Reviewer #2:
(1) Introduction – can you add some literature about general insect morphology? Not every reader knows insect morphology.
Response: Thank you for your suggestion. Due to spatial constraints, we were unable to extensively cover the literature on mouthpart morphology in the introduction. However, in subsequent discussion, we thoroughly explores the variations in mouthpart morphology among insects with diverse diets and delves into the functional aspects of different components of their mouthparts. We have cited more relevant references in this section.
(2) line 90: “Scanning electron microscopy - cope (SU8010, Hitachi, Japan) was used to observe and take photos at 5KV – there should be the equipment, not the technology.
Response: Thank you for your suggestion. This sentence has been modified as suggested in line 94.
(3) Results – line 107-108 – there is a reference to Fig. 1c and 1d, but not for 1a and 1b. These figures are also described below (line 111) as Adult with capital letter, which is not needed, lowercase letters are sufficient.
Response: Thank you for your suggestion. Fig. 1a and 1b have been cited and this sentence has been modified as suggested in line 117.
(4) line 115: Types and morphology would be more precise.
Response: Thank you for your suggestion. This sentence has been modified as suggested in line 121.
(5) line 116: “According to the scanning electron microscopy (…)” SEM cannot suggest anything being machine – I suggest to change for: “After analyzing under SEM it is (…)”.
Response: Thank you for your suggestion. This sentence has been rewritten as “According to the scanning electron micrographs” in line 122.
(6) line 120: shortcut Sty (for s. styloconica)– the abbreviation SS is most common in the literature and I suggest to use it in this article.
Response: Thank you for your suggestion. In many other literatures, Sty has also been found as the shortcut for sensilla styloconica. This abbreviation gives a clearer identification of this kind of sensilla.
(7) line 156: “There sockets are convex with some globular process around (Figure. 2g, h and 3b)”. The convex socket is hardly visible, can you provide one bette3r shot for this element?
Response: Thank you for your suggestion. This type of sensilla is common in ladybirds and often situated at the same position. Their morphology was similar in different species. In fact, the convex socket is not very obvious as Figure 2g, while the round protrusion are obvious. In other species, these protrusion maybe situated on the high stylus, so they can be considered as a part of the socket.
(8) line 172: can you provide better figure for Sp2? In the Fig. 3b there is some trace/imprint of poor quality, easy to confuse with anything else.
Response: Thank you for your suggestion. Sp2 has been renamed as Sp as suggested. The magnification of Sp can be seen in Figure 3d. In the process of observation under SEM, we have magnified Sp2 to see more details, but no pore or other structure has been found. The surface of Sp is similar to the cuticle around. Finally, we chose this picture because all three types of sensilla on it are quite clear.
(9) line 232: lack of a verb: Prostheca dorsally (has/possess?) spiny processes, ventrally multi-layered slender bristles, ventrally 232 from the molar to the abdomen.
Response: Thank you for your suggestion. This sentence has been rewritten as suggested in line 245-246.
(10) line 254: too long sentence: “The lacinia bears setae and spines 253 on its margin and at its apex. and is closely located below the galea, and a row of Sb1 are 254 arranged at the edge of lacinia, dorsal smooth, ventral distribution of Sch1, Sb1, Sp1, Bb, 255 Cp (Figure. 6d, e, f)”. What about the dot after the apex?
Response: Thank you for your suggestion. This sentence has been rewritten as suggested in line 266-267.
(11) line 274: The Cp is not marked on the photo 6h, i - is it a mistake?
Response: Thank you for your suggestion. Cp is universal in the sensory region on the top of maxillary palpi, while because they are too small, the magnification needs to be large enough in order to see them clearly. Figure 6h and 6i was used to show Sb2, Sty1 and Sty2, and this photo is not large enough to see Cp clearly.
(12) line 280: ligula should be with a capital letter, and also sensilla (Sch1, Sb1, Cp) are not shown – can you provide such figure?
Response: Thank you for your suggestion. This sentence has been modified as suggested. The figure of Sch1 and Cp has been added as Figure 7d as suggested.
(13) line 282: “The ventral surface of labial palp is completely covered by epidermal processes, including water droplet processes, conical processes, and hairy processes (Figure. 7d, e)” – you wrote about the ventral surface of Lp, while Fig. 7d, e are about Lig and Prmt and also there is lack of any described processes marked.
Response: Thank you for your suggestion. This sentence has been modified in detail as suggested in line 267-268. The cuticular processes are not the main object in this text, so we have not described them in detail.
(14) lines 288-289 – there is a lack of Sp2 – it is in the Fig. 7b.
Response: Thank you for your suggestion. This Figure has been changed and explained in detail as suggested in line 299-300.
(15) line 328: there is some information about prostheca – can you move general info about it to the beginning of the article?
Response: Thank you for your suggestion. Due to spatial constraints, we were unable to add more infomation about prostheca in the introduction. However, in subsequent discussion, we thoroughly discuss the function of prostheca and the difference in their morphology. We have cited more relevant references in this section.
(16) line 384: Nitidula carnaria – the specific name should be with lowercase letter.
Response: Thank you for your suggestion. This sentence has been modified as suggested in line 399.
Reviewer 3 Report
Comments and Suggestions for Authors
Please see the attached file with my review (comments and sugestions). Thank you.
Sincerely,

The English is acceptable.
Author Response
Reviewer #3:Review Final Opinion: In my view, the “Manuscript ID: insects-2791069” presents significant scientific collaboration within the scopes of Electron Microscopy, Insect Morphology, and Morphofunctional Study of Insect Microstructures. It features clear and concise writing, along with high-quality electron micrographs.
(1) However, throughout the text, the authors missed an opportunity to enhance their work by not citing other Western authors who had previously explored a similar idea. These authors had related the morphology of mouthparts to their functioning and feeding behavior during the feeding process. For instance, consider the recent Scanning Electron Microscopy (SEM) study on the mouthparts of Haematobia irritans, a hematophagous muscomorph Diptera species. In that study, insects were captured and immediately fixed in the field during their hematophagic activity on the host. This allowed researchers to highlight the functioning of microstructures associated with hematophagous feeding. I believe that if a prior publication inspires new research, even within a different insect order, it is just and noble to cite that initial article as the first reference in the new publication. Furthermore, I advocate for Western authors to cite Eastern authors and vice versa, as commonly seen in high-impact journal articles. Science, by its intrinsic nature, is universal, free, and continuously evolving. It draws inspiration from results produced by authors from all corners of the world, regardless of ethnicity, culture, or politics, for the betterment of scientific progress.
Response: Thank you very much for your suggestion. We fully agree with your statement that more authoritative Western literature should be referenced to enhance our work and promote cultural exchange between the East and the West. Science knows no boundary. Science, by its intrinsic nature, is universal, free, and continuously evolving. It draws inspiration from results produced by authors from all corners of the world, regardless of ethnicity, culture, or politics, for the betterment of scientific progress. We have already cited the literature you mentioned in line 41-46, and in our future research, we will also pay attention to similar issues and learn from outstanding Western scholars.
(2) As another suggestion, the ‘Cuticular pore (Cp)’ presented by the authors in Figure 3H could be illustrated through Transmission Electron Microscopy (TEM). If it is not possible to do this in the manuscript, I recommend that the authors at least do not present it immediately after Figure 3G (böhn bristle), and instead cite a publication that shows a cross-section of the Cp using TEM. I suggest this to prevent readers from confusing the Cp with wall pores, which are structures with different morphology and functions.
Response: Thank you for your suggestion. Cp has been classified as glandular structure with Pp, and their positon in Table1 has been changed as suggested. In discussion, we have cited literature that contain study of TEM to discuss their function. These sentences have been rewritten as suggested in line 129, 211-213, 403-409.
(3) I also noticed that in Figure 4H, only ‘palmate processes’ are shown and classified, which have finger-like projections. However, upon closer examination of the figure, I observed at least two other types of microstructures/sensilla. One appears to have a single articulated projection, and the other has two projections. I recommend pointin them out in the figure and distinguishing them. If the authors conduct a broader literature review on sensilla from other insect orders, they may find similar microstructures that have already been classified and typologically named.
Response: Thank you for your suggestion. The morphological differences of cuticular processes are quite significant, and we have only classified them based on their characteristics. However, each type of cuticular processes also exhibits subtle variations in morphology at different locations, such as the difference in branch numbers as you mentioned. We have provided corresponding explanation in the text in line 223-225.
(4) Despite these suggestions, I consider the manuscript suitable for publication in the Insects journal, provided that the authors address the few but important improvements suggested.
Response: Thank you for your suggestion.
Round 2
Reviewer 1 Report
Comments and Suggestions for Authors
The authors reacted favourably to the requests of the reviewer.
Some mistakes remain (see attached file)

Author Response
Thank you for your suggestion. We have carefully studied the valuable comments and revised the paper accordingly. Two literatures have been added into the manuscript and all the errors have been modified as suggested. Please see the attachment.
